# Structural basis of ligand recognition and design of antihistamines targeting histamine H₄ receptor

Ruixue Xia[1,6], Shuang Shi [2,6], Zhenmei Xu[1], Henry F. Vischer [2], Albert D. Windhorst[3], Yu Qian[1], Yaning Duan[1], Jiale Liang[1], Kai Chen[1], Anqi Zhang[4], Changyou Guo[4], Rob Leurs [2] ✉ & Yuanzheng He [1,5] ✉

The histamine H₄ receptor (H₄R) plays key role in immune cell function and is a highly valued target for treating allergic and inflammatory diseases. However, structural information of H₄R remains elusive. Here, we report four cryo-EM structures of H₄R/G_i complexes, with either histamine or synthetic agonists clobenpropit, VUF6884 and clozapine bound. Combined with mutagenesis, ligand binding and functional assays, the structural data reveal a distinct ligand binding mode where D94[3.32] and a π-π network determine the orientation of the positively charged group of ligands, while E182[5.46], located at the opposite end of the ligand binding pocket, plays a key role in regulating receptor activity. The structural insight into H₄R ligand binding allows us to identify mutants at E182[5.46] for which the agonist clobenpropit acts as an inverse agonist and to correctly predict inverse agonism of a closely related analog with nanomolar potency. Together with the findings regarding receptor activation and G_i engagement, we establish a framework for understanding H₄R signaling and provide a rational basis for designing novel antihistamines targeting H₄R.

Histamine, a biogenic amine chemical messenger, plays pivotal roles in various physiological and pathophysiological processes through binding and activating histamine receptors (H₁R-H₄R), members of the G-protein coupled receptors (GPCRs) superfamily[1,2]. H₁R is the first identified histamine receptor and is closely linked to allergic reactions in humans. H₁R is widely expressed throughout the body and primarily signals through the G_q pathway[1]. It has been successfully targeted for allergic disorders, leading to the development of a range of blockbuster drugs for alleviating allergic symptoms. Similarly, H₂R is also widely expressed in the body and predominantly signals through the G_s pathway[1]. H₂R has emerged as a successful target for blockbuster

antihistamine drugs, revolutionizing the treatment of gastric acid-related conditions such as stomach ulcers.

On the other hand, H₃R is mainly expressed in the central nervous system and is involved in e.g. cognitive functions, sleep-wake regulation, and energy homeostasis[3]. In 2014, pitolisant (Wakix®), an inverse agonist of H₃R, received approval for the treatment of refractory narcolepsy[4]. Following the complete sequencing of the human genome, the H₄R is the most recently identified histamine receptor. It is mainly expressed in immune cells including eosinophils, T cells, dendritic cells, basophils, and mast cells[1,5]. H₄R plays a crucial role in mediating immune cell migration, cytokine release, and IL-17 production. Notably, H₄R is specifically associated with prevailing

[1]Laboratory of Receptor Structure and Signaling, HIT Center for Life Sciences, School of Life Science and Technology, Harbin Institute of Technology, Harbin, China. [2]Department of Medicinal Chemistry, Amsterdam Institute for Molecular Life Sciences, Faculty of Science, Vrije Universiteit Amsterdam, De Boelelaan 1108, 1081 HV Amsterdam, The Netherlands. [3]Department of Radiology and Nuclear Medicine, VU University Medical Center Amsterdam, Amsterdam, The Netherlands. [4]School of Life Science and Technology, Harbin Institute of Technology, Harbin, China. [5]Frontiers Science Center for Matter Behave in Space Environment, Harbin Institute of Technology, Harbin, China. [6]These authors contributed equally: Ruixue Xia, Shuang Shi. ✉e-mail: r.leurs@vu.nl; ajian.he@hit.edu.cn

inflammatory conditions such as psoriasis, atopic dermatitis, asthma, inflammatory bowel disease (IBD), and arthritis. Due to its significant impact on immune cells, $H_4R$ is seen as an attractive target for the treatment of these inflammatory diseases. Both $H_3R$ and $H_4R$ predominantly couple with $G_i$/o proteins[1].

The successes of therapeutic targeting $H_1R$, $H_2R$, and $H_3R$ have inspired and fueled the field to develop numerous $H_4R$ antagonists and agonists for associated diseases. JNJ7777120 was first generated as a selective antagonist for $H_4R$ based on its ability to inhibit histamine-induced $G_i$ activation[6]. It exhibits 1000-fold selectivity over $H_1R$, $H_2R$, and $H_3R$. JNJ7777120 blocks histamine-induced chemotaxis in mouse mast cells and neutrophil infiltration, suggesting its potential for the treatment of inflammatory diseases. Later, JNJ7777120 was discovered to have the ability to recruit β-arrestin without activating G proteins[7], i.e. being a biased agonist for $H_4R$[8], offering a promising approach to minimize undesired side effects. Several compounds targeting $H_4R$ have entered clinical trials for asthma, IBD, and arthritis, but so far not one has prevailed[9].

In contrast to $H_1R$, $H_2R$, and $H_3R$, structural information for $H_4R$ is currently absent. The first reported histamine receptor structure is the x-ray structure of antagonist doxepin-bound $H_1R$ solved in 2011[10]. The structure reveals the overall framework of $H_1R$ and the mode of antagonist binding which serves as a template for designing and developing $H_1R$ antihistamines. It took a decade for the active cryo-EM structure of $H_1R$ to appear[11]. The active structure reveals a shrinkage of ligand binding pocket size via the ionic interaction of histamine with key residues in transmembrane helix 3, 6, and 7 (TM3,6 and 7), followed by an outward movement of TM6 to open the intracellular cavity for the $G_q$-protein to engage the GPCR, as the main mechanism for $H_1R$ activation. Later, a cryo-electron microscopy (cryo-EM) structure of antagonist famotidine-bound $H_2R$ via a fusion strategy[12] and a crystal structure of antagonist PF03654746-bound $H_3R$[13] were subsequently reported, revealing the inactive conformations of $H_2R$ and $H_3R$. Most recently, the cryo-EM structures of apo and antihistamine-bound $H_1R$ were reported[14].

In this work, we resolve the cryo-EM structures of $H_4R$/$G_i$ complexes bound with the endogenous ligand histamine, and synthetic agonists clobenpropit, VUF6884, and clozapine. Through a combination of ligand binding experiments and functional assays, we reveal a distinctive ligand binding mode of $H_4R$ that significantly differs from that of $H_1R$. We further uncover the mechanism of receptor activation and $G_i$ coupling. The information unveiled by our study is not only essential for the molecular understanding of $H_4R$ signaling but also for the development of novel compounds that target $H_4R$ to treat inflammatory diseases.

## Results

### Overall architecture of $H_4R$/$G_i$ complex

We co-expressed human $H_4R$ with $G_{i1}$ protein, together with antibody fragment scFv16 that specifically recognizes the N-terminus of $Gα_{i1}$, in *Spodoptera frugiperda* (*Sf9*) insect cells and purified the complex by a conventional membrane protein purification method of our lab[11] (Supplementary Fig. 1, for details see methods). We solved the $H_4R$/$G_i$ complex bound with histamine, clobenpropit, VUF6884 and clozapine at resolutions of 3.07 Å, 3.06 Å, 3.01 Å and 3.21 Å, respectively, by the gold standard of FSC = 0.143 (Table 1 and Supplementary Fig. 2). The overall structures of the $H_4R$/$G_i$ complexes closely resemble the conventional GPCR/G-protein complex, where the receptor/G-protein interaction primarily occurs through the Gα subunit of $G_i$ (Fig. 1). Local resolution analysis shows that the core of the Gβ subunit, the transmembrane domains of the GPCR and the $G_i$ interface of scFv16 have the highest resolution. The alpha-helical domain (AHD) of $Gα_i$ is missing from the density map due to its high flexibility in the nucleotide-free state. The good density map of the receptor allows us to unequivocally assign residue 14–372 while lacking amino acids 204–292 of the intracellular loop 3 (ICL3) due to the high flexibility of this

**Table 1 | Cryo-EM data collection and refinement statistics**

| | $H_4R$/Hista-mine/$G_i$ | $H_4R$/Cloben-propit/$G_i$ | $H_4R$/VUF6884/$G_i$ | $H_4R$/Cloza-pine/$G_i$ |
|---|---|---|---|---|
| | EMD-36712 | EMD-36716 | EMD-36715 | EMD-36714 |
| | 8JXT | 8JXX | 8JXW | 8JXV |
| **Data collection and processing** | | | | |
| Magnification | 130,000 | 130,000 | 130,000 | 64,000 |
| Voltage (kV) | 300 | 300 | 300 | 300 |
| Electron exposure (e⁻/Å²) | 60 | 60 | 60 | 50 |
| Defocus range (μm) | 1.2–2.2 | 1.2–2.2 | 1.2–2.2 | 1.8 |
| Pixel size (Å) | 1.1 | 1.1 | 1.1 | 1.08 |
| Symmetry imposed | C1 | C1 | C1 | C1 |
| Initial particle image (no.) | 1.9 M | 1.0 M | 1.7 M | 1.5 M |
| Final particle image (no.) | 340k | 86k | 334k | 106k |
| Map resolution (Å) | 3.07 | 3.06 | 3.01 | 3.21 |
| FSC threshold | 0.143 | 0.143 | 0.143 | 0.143 |
| **Refinement** | | | | |
| Initial model used (PDB code) | AlphaFold-Q9H3N8-v1 | AlphaFold-Q9H3N8-v1 | AlphaFold-Q9H3N8-v1 | AlphaFold-Q9H3N8-v1 |
| Model Resolution (Å) | NA | NA | NA | NA |
| Map sharpening *B* factor (Å²) | –129.4 | –91.6 | –107.4 | –116.7 |
| **Model composition** | | | | |
| Non-hydrogen atoms | 8294 | 8385 | 8389 | 7935 |
| Protein residues | 1107 | 1108 | 1109 | 1040 |
| Ligands | 1 | 1 | 1 | 1 |
| ***B* factor (Å²)** | | | | |
| Protein | 68.27 | 80.03 | 76.63 | 56.73 |
| Ligand | 0 | 86.36 | 78.07 | 30.06 |
| **R.m.s. deviations** | | | | |
| Bond length (Å) | 0.005 | 0.006 | 0.006 | 0.006 |
| Bond angles (°) | 1.044 | 0.746 | 0.795 | 0.78 |
| **Validation** | | | | |
| MolProbity score | 1.86 | 1.61 | 1.64 | 2 |
| Clashscore | 7.01 | 5.09 | 6.24 | 12.39 |
| Poor rotamers (%) | 0 | 0 | 0 | 0 |
| **Ramachandran plot** | | | | |
| Favored (%) | 94.58 | 95.13 | 95.77 | 94.19 |
| Allowed (%) | 5.42 | 4.87 | 4.23 | 5.81 |
| Disallowed (%) | 0 | 0 | 0 | 0 |

region. Of note, the clozapine-bound $H_4R$/$G_i$ complex has a slightly lower resolution than the histamine-, clobenpropit- and VUF6884-bound $H_4R$/$G_i$ complexes.

### Histamine binding

The natural ligand, histamine, is well resolved in the ligand binding pocket of $H_4R$, surrounded by D94[3.32], Y95[3.33], C98[3.36], Q347[7.42], Y319[6.51], F344[7.39], and W348[7.43]. Histamine establishes direct polar interaction

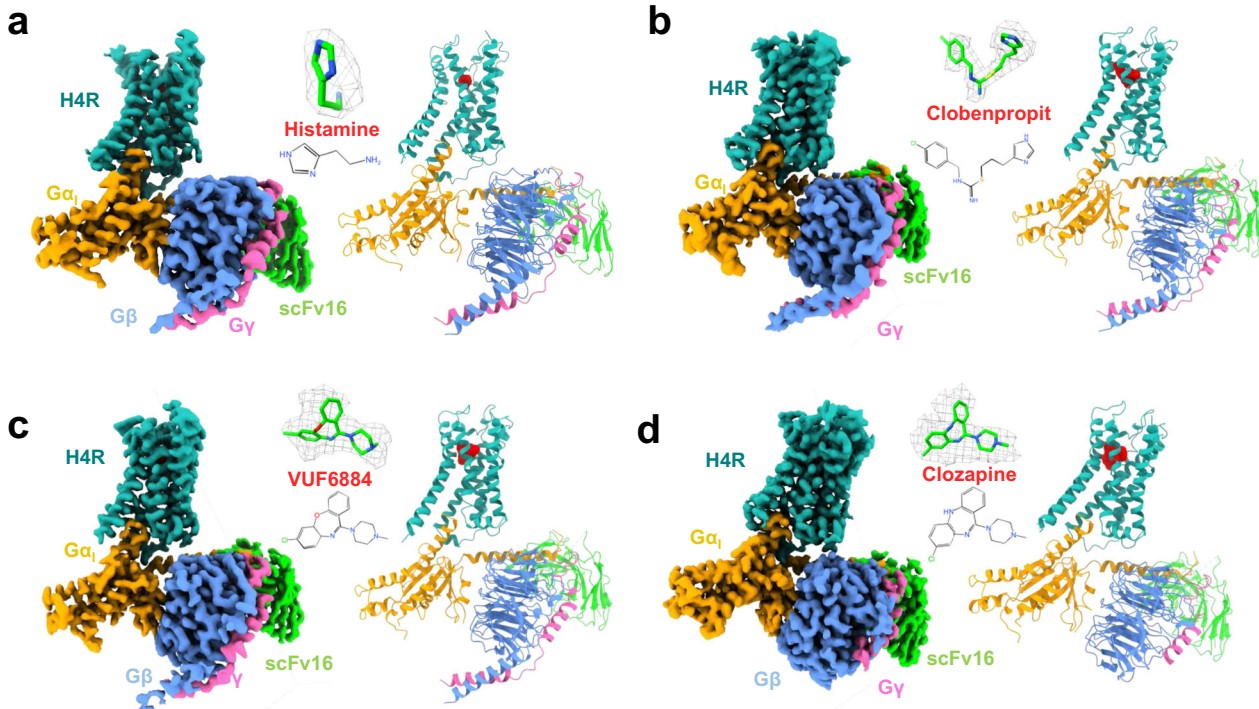

**Fig. 1 | The overall structures of H₄R/Gᵢ complexes. a–d** Histamine-bound H₄R/Gᵢ, Clobenpropit-bound H₄R/Gᵢ, VUF6884-bound H₄R/Gᵢ, and Clozapine-bound H₄R/Gᵢ, respectively. Left panel, orthogonal views of the cryo-EM density map; right panel, model of the complex in the same view and color scheme as shown in the left panel. Ligands, histamine, Clobenpropit, VUF6884, and Clozapine were shown in a stick model with a density map (contour level of 0.4 in chimera) and in actual chemical structure in the middle of each sub-figure.

with D94[3.32], Y95[3.33], and F344[7.39] (Fig. 2a, b). When bound to the receptor, histamine can act as a dication molecule where both the imidazole ring and the amine of the tails are charged[15]. The positively charged imidazole ring is positioned such that next to an interaction with D94[3.32], it also forms a cation-π interaction with F344[7.39]. We have additionally observed a small density adjacent to E182[5.46], which is insufficient to accommodate a histamine molecule. However, based on its abundance in cells, we assigned this density as a phosphate ion, effectively linking the positively charged primary amine of histamine to E182[5.46] through hydrogen-bond interaction. An alignment of the key residues of the pocket among the histamine receptor family shows that the pocket is more conserved in H₃R and H₄R than in H₁R and H₂R (Fig. 2c). Much to our surprise, histamine uses different strategies to engage H₁R and H₄R. In binding to H₁R, the imidazole ring is orientated to TM5 and TM3, forming direct polar interactions with N198[5.46] and T112[3.37]; in contrast, for H₄R binding, the imidazole ring takes a completely opposite direction and is orientated to TM7 to interact with F344[7.39] (Fig. 2d and Supplementary Fig. 3c). A key difference of the interaction network is the involvement of N198[5.46] for histamine binding in H₁R, while E182[5.46] is distant from histamine and does not directly engage the ligand in H₄R. A comparison of monoamine ligand binding modes shows that dopamine, adrenaline, and serotonin use identical modes as histamine in H₁R to engage their respective GPCRs, while histamine in H₄R uses a completely opposite binding orientation (Fig. 2d and Supplementary Fig. 3e, f). Despite the engaging difference in binding modes, all monoamines form key polar interactions with the conserved D[3.32], highlighting the well-accepted importance of this residue in monoamine ligand binding.

We used molecular dynamic (MD) simulations to examine the stability of histamine binding in the pocket of H₄R. Triplicate 200 ns runs show that histamine and the anion phosphate are very stable during the simulations (Supplementary Fig. 4c, d and Supplementary Table 1 and Supplementary Movie 1). A closer examination of a snapshot from the MD simulations reveals that the phosphate molecule sits at the gap between E182 and Y318, acting as a barrier that prevents histamine from escaping the cage formed by Y318, F344, W348, D94, and Y95 (Supplementary Fig. 4c). This arrangement leads to a highly stable histamine binding pose, as evidenced by minimal changes during the simulation (left upper panel of Supplementary Fig. 4d), further supported by the straight line from the RMSD analysis (left lower panel of Supplementary Fig. 4d). In contrast, simulations without the phosphate result in histamine flipping around in the binding pocket, as depicted in the snapshots of the simulation (Supplementary Fig. 4d, right upper panel). This dynamic behavior is reflected in the substantial fluctuation of the RMSD curves (the lower right panel of Supplementary Fig. 4d). Collectively, these findings suggest that the phosphate anion plays a crucial role in stabilizing histamine binding.

We employed radioligand binding experiments to investigate the contribution to ligand binding of each residue of the H₄R ligand binding pocket. In [³H]histamine binding assays, none of the mutants bind to the radioligand with sufficiently high affinity for a precise evaluation of the contribution of each residue (Fig. 2e), while all mutants have a similar surface expression level compared to the wild-type H₄R, as measured by an anti-HA ELISA (Supplementary Fig. 7b). We then custom-synthesized [³H]JNJ7777120, a highly selective and potent antagonist of H₄R. The measured affinities (pKᵢ) of all the examined ligands, determined through displacement of [³H]JNJ7777120, are 7.7, 8.0, 7.7, and 6.5 for histamine, clobenpropit, VUF6884, and clozapine, respectively (Table 2). These values align closely with the reported pKᵢ values of 7.7, 7.9, 7.6, and 6.4 for histamine, clobenpropit, VUF6884, and clozapine, respectively[16,17]. Importantly, except for D94[3.32] and W348[7.43], most H₄R mutants maintain a substantial affinity for [³H]JNJ7777120 (Fig. 2e; Supplementary Figs. 3i and 5a), enabling a thorough assessment of the role played by each amino acid in ligand binding. The lack of [³H]JNJ7777120 binding

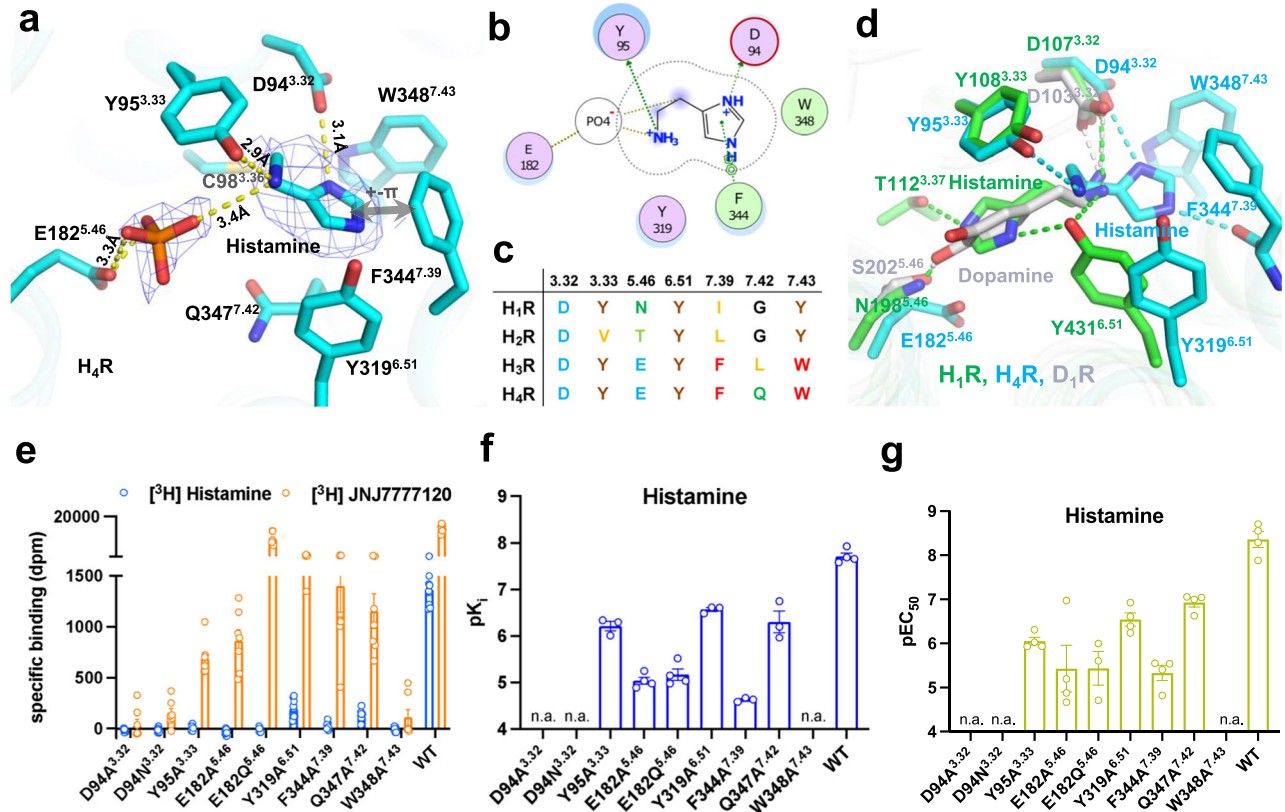

**Fig. 2 | Histamine recognition and binding of H4R. a** The ligand binding pocket of histamine. **b** A schematic map of histamine/receptor interaction. Green color, hydrophobic interaction; purple color, polar interaction. **c** Conservation of key residues of the ligand binding pocket among histamine receptor family. **d** A comparison of histamine binding between H4R and H1R (PDB:7dfl). **e** [³H] histamine and [³H] JNJ7777120 binding for H4R mutants. **f** pKi of histamine binding of H4R mutants. **g** pEC50 of Gi activation of H4R mutants by histamine. From **e** to **g**, data are presented as mean values ± SEM.; $n = 4-9$ independent experiments for **e**, $n = 3-4$ independent experiments for **f**, and $n = 3-4$ independent experiments for **g**. Each point in the figure represents an independent experiment. Source data are provided as a Source Data file.

by the D94A[3.32] and D94N[3.32] mutants implicates that the conserved D94[3.32] in TM3 also plays a crucial role in JNJ7777120 binding. This matches well with docking results for JNJ7777120 where D94[3.32] forms a key salt-bridge interaction with the amine of the methylpiperazine ring of JNJ7777120 (Supplementary Fig. 3g, h). In [³H]JNJ7777120-histamine competition binding experiments, F344A[7.39] shows a dramatic decrease of histamine binding, consistent with the structural observation of a direct interaction between the imidazole ring with F344[7.39]. In addition, the Y95A[3.33], E182A[5.46], E182Q[5.46], Y319A[6.51], and Q347A[7.42] mutants all show a substantial decrease of the pKi value of histamine (Fig. 2f and Supplementary Fig. 5b). We also used a BRET-based Gi-protein activation assay[18] to evaluate the contribution of each key pocket residue on receptor activation. In agreement with the pKi binding data, G-protein activation by histamine is completely abrogated by the D94[3.32], D94N[3.32], and W348A[7.43] mutations, while F344A[7.39] severely decreases receptor activation and all other mutants cause a substantial loss of H4R activation (Fig. 2g and Supplementary Fig. 8a).

**Binding of synthetic H4R agonists**
We also evaluated the binding modes of three synthetic H4R agonists. The overall structures of histamine-, clobenpropit-, VUF6884- and clozapine-bound H4R are almost identical (Supplementary Fig. 3b) with a root mean square deviation (r.m.s.d.) of 0.299 Å over 215 pairs of Cα. While histamine only occupies half of the ligand binding pocket on the TM7 side, the synthetic agonists clobenpropit, VUF6884, and clozapine cover the whole orthosteric pocket (Supplementary Fig. 3a). Clobenpropit is a highly potent antagonist/inverse agonist of H3R and

a partial agonist of H4R[19,20]. In the clobenpropit-bound H4R complex, its imidazole ring interacts with D94[3.32]; on the other side, E182[5.46] forms crucial salt-bridge interactions with both the N8 and N10 atom of the isothiourea group of clobenpropit (Fig. 3a, b). In addition, T99[3.37] also forms a polar interaction with the N10 atom of clobenpropit. Interestingly, like for histamine the positively charged imidazole ring of clobenpropit forms cation-π interactions with a π-π network formed by F344[7.39], W348[7.43], and Y319[6.51] (Figs. 2a and 3a; and Supplementary Fig. 3d). In support of this, mutation of D94[3.32] completely abrogates receptor activity and mutation of F344[7.39], Y319[6.51], and W348[7.43] strongly inhibit ligand binding (Fig. 3c; Supplementary Figs. 6a, b and 7a) and receptor activation (Fig. 3d; Supplementary Figs. 8b and 9a, b).

Clozapine, an atypical antipsychotic medication approved by the FDA for the treatment of schizophrenia, functions as an antagonist of dopamine D4 receptor[21,22]. It is also a multi-target drug and binds with moderate to high affinity to a fair number of aminergic receptors, including serotonin 5-HT2A/2C receptor, H1R, and H4R. Interestingly, clozapine has been found to activate H4R, which might be related to the known side effect of agranulocytosis by clozapine[20,23]. VUF6884 is a more potent analog of clozapine at H4R[24] and only differs from clozapine by the position of the chlorine atom and a substitution of the nitrogen atom with an oxygen atom at the dibenzodiazepine ring (Fig. 3b). In line with their structural similarity, both compounds exhibit a similar binding mode when interacting with H4R (Fig. 3a). The positively charged methyl-1-piperazinyl group forms a direct ionic interaction with D94[3.32] and is positioned to the π-π network formed by F344[7.39], W348[7.43], and Y319[6.51], resembling the H4R interaction of the

## Table 2 | Activity of ligands at H₄R and selected mutants

| hH₄R | Histamine pK$_i$ | Fold (pK$_i$) | pEC$_{50}$ | Fold (pEC$_{50}$) | Clobenpropit pK$_i$ | Fold (pK$_i$) | pEC$_{50}$ | Fold (pEC$_{50}$) | Clozapine pK$_i$ | Fold (pK$_i$) | pEC$_{50}$ | Fold (pEC$_{50}$) | VUF6884 pK$_i$ | Fold (pK$_i$) | pEC$_{50}$ | Fold (pEC$_{50}$) | JNJ7777120 pK$_i$ | Fold (pK$_i$) |
|---|---|---|---|---|---|---|---|---|---|---|---|---|---|---|---|---|---|---|
| WT | 7.7 ± 0.1 (4) | | 8.4 ± 0.2 (4) | | 8.0 ± 0.1 (3) | | 8.1 ± 0.2 (6) | | 6.5 ± 0.0 (4) | | 6.6 ± 0.2 (4) | | 7.7 ± 0.1 (4) | | 7.6 ± 0.1 (4) | | 8.5 ± 0.1 (3) | |
| D94³·³²A | ND (3) | | ND (4) | | ND (3) | | ND (3) | | ND (3) | | ND (3) | | ND (3) | | ND (4) | | ND (3) | |
| D94³·³²N | ND (3) | | ND (4) | | ND (3) | | ND (3) | | ND (3) | | ND (3) | | ND (3) | | ND (4) | | ND (3) | |
| Y95³·³³A | 6.2 ± 0.1 (3) **** p < 0.0001 | 32 ↓ | 6.1 ± 0.1 (4) **** p < 0.0001 | 200 ↓ | 7.3 ± 0.2 (3) ** p = 0.0013 | 5 ↓ | ND (3) | | 5.7 ± 0.2 (4) **** p < 0.0001 | 6 ↓ | ND (4) | | 6.6 ± 0.2 (4) **** p < 0.0001 | 12 ↓ | 6.6 ± 0.2 (4) ** p = 0.0036 | 10 ↓ | 6.7 ± 0.2 (3) **** p < 0.0001 | 63 ↓ |
| E182⁵·⁴⁶A | 5.0 ± 0.1 (4) **** p < 0.0001 | 501 ↓ | 5.4 ± 0.5 (4) **** p < 0.0001 | 1000 ↓ | 6.8 ± 0.0 (3) **** p < 0.0001 | 16 ↓ | 6.1 ± 0.3 (7) inverse agonism | 100 ↓ | 6.2 ± 0.1 (4) * p = 0.0385 | 2 ↓ | 6.4 ± 0.3 (3) ns p = 0.9475 | 2 ↓ | 7.0 ± 0.2 (4) ** p = 0.0086 | 5 ↓ | 7.6 ± 0.3 (4) ns p = 0.9999 | 1 ↓ | 6.0 ± 0.3 (3) **** p < 0.0001 | 316 ↓ |
| E182⁵·⁴⁶Q | 5.2 ± 0.1 (4) **** p < 0.0001 | 316 ↓ | 5.4 ± 0.4 (3) **** p < 0.0001 | 1000 ↓ | 7.2 ± 0.0 (3) *** p = 0.0004 | 6 ↓ | 8.0 ± 0.3 (3) inverse agonism | 1 ↓ | 5.9 ± 0.0 (4) **** p < 0.0001 | 4 ↓ | 6.5 ± 0.2 (4) ns p = 0.9969 | 1 ↓ | 6.9 ± 0.0 (4) ** p = 0.0021 | 6 ↓ | 7.4 ± 0.1 (4) ns p = 0.9351 | 2 ↓ | 7.4 ± 0.1 (3) ** p = 0.0038 | 13 ↓ |
| Y319⁶·⁵¹A | 6.6 ± 0.0 (3) **** p < 0.0001 | 13 ↓ | 6.5 ± 0.2 (4) *** p = 0.0004 | 80 ↓ | 7.1 ± 0.0 (3) *** p < 0.0001 | 8 ↓ | ND (4) | | 6.7 ± 0.0 (4) ns p = 0.1629 | 2 ↑ | 6.4 ± 0.1 (3) ns p = 0.9125 | 2 ↓ | 7.6 ± 0.1 (4) ns p = 0.9845 | 1 ↓ | 7.2 ± 0.2 (4) ns p = 0.6146 | 3 ↓ | 7.0 ± 0.0 (3) *** p = 0.0002 | 32 ↓ |
| F344⁷·³⁹A | 4.6 ± 0.0 (3) **** p < 0.0001 | 1259 ↓ | 5.3 ± 0.2 (4) **** p < 0.0001 | 1259 ↓ | 5.9 ± 0.1 (3) **** p < 0.0001 | 126 ↓ | 6.0 ± 0.3 (4) *** p = 0.0005 | 126 ↓ | 5.8 ± 0.0 (4) **** p < 0.0001 | 5 ↓ | 6.1 ± 0.1 (4) ns p = 0.2316 | 3 ↓ | 6.7 ± 0.1 (4) *** p = 0.0003 | 10 ↓ | 6.7 ± 0.1 (4) * p = 0.0121 | 8 ↓ | 6.7 ± 0.2 (3) **** p < 0.0001 | 63 ↓ |
| Q347⁷·⁴²A | 6.3 ± 0.2 (3) **** p < 0.0001 | 25 ↓ | 6.9 ± 0.1 (4) ** p = 0.0048 | 32 ↓ | 7.8 ± 0.2 (3) ns p = 0.4132 | 2 ↓ | 9.2 ± 0.2 (4) ns p = 0.0872 | 13 ↑ | 7.7 ± 0.1 (4) **** p < 0.0001 | 16 ↑ | 8.4 ± 0.1 (4) **** p < 0.0001 | 63 ↑ | 7.3 ± 0.2 (4) ns p = 0.2886 | 3 ↓ | 8.1 ± 0.2 (4) ns p = 0.1077 | 3 ↑ | 6.6 ± 0.2 (3) **** p < 0.0001 | 80 ↓ |
| W348⁷·⁴³A | ND (3) | | ND (4) | | ND (3) | | ND (3) | | ND (3) | | ND (3) | | ND (3) | | ND (4) | | ND (3) | |

Data were shown as mean ± sem of at least three independent experiments which were performed in duplicate. Binding affinity (pK$_i$) was determined with [³H] JNJ7777120 displacement assay. Potency (pEC$_{50}$) was determined with a BRET-based Gi activation biosensor. Fold decrease (↓) or increase (↑) in binding affinity or potency compared to WT H₄R is indicated. Statistical difference ($p < 0.05$) in pK$_i$ or pEC$_{50}$ for the mutants in comparison to WT H₄R was analyzed using One-way AVONA followed by Dunnett's multiple comparison test. Statistical differences are indicated with asterisk and corresponding $p$-values are shown in roman. ns not significant. ND not detectable.

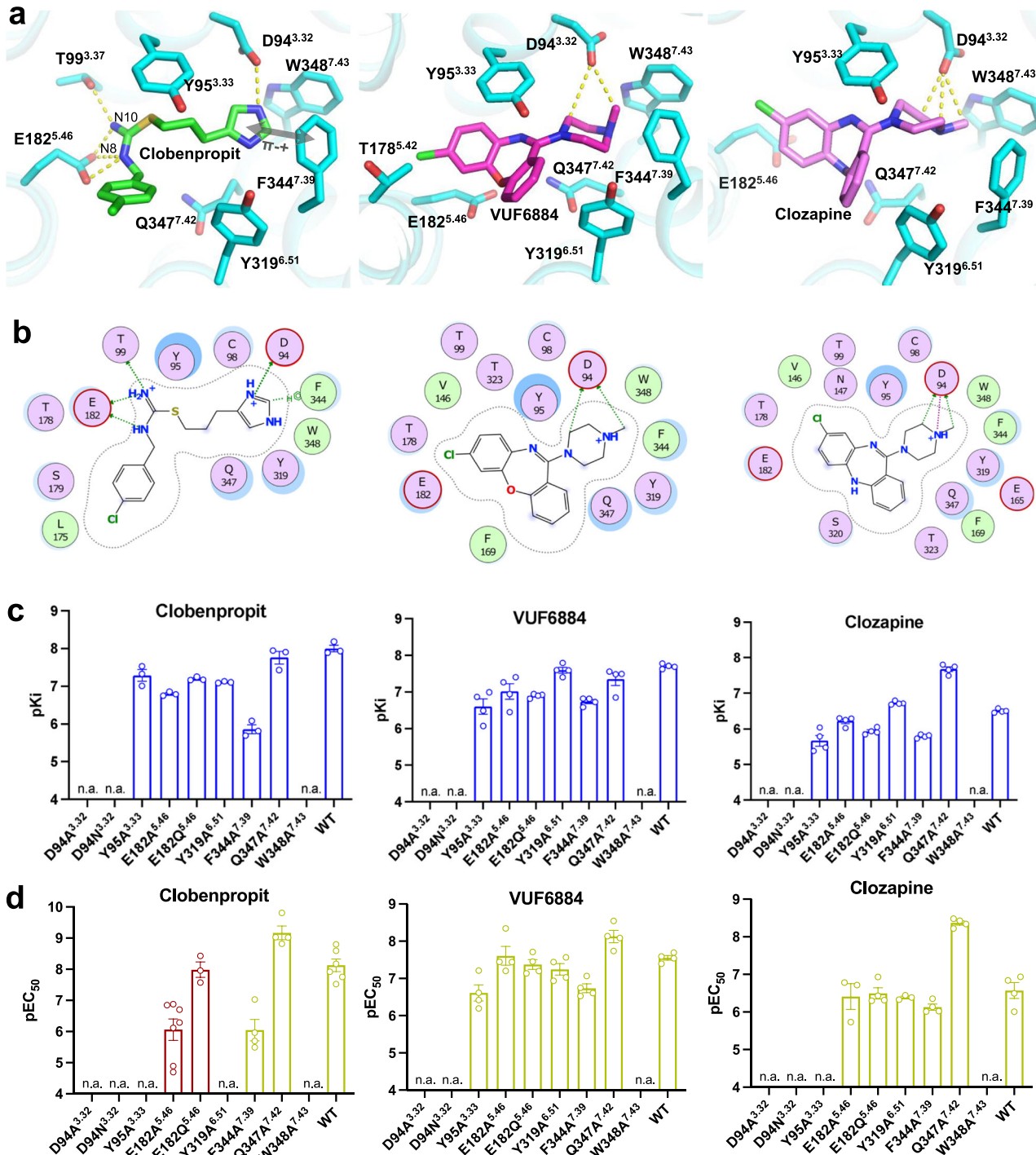

**Fig. 3 | Non-histamine agonists recognition and binding of H₄R. a** The ligand binding pocket of Clobenpropit, VUF6884, and Clozapine. **b** A schematic map of Clobenpropit, VUF6884, and Clozapine/receptor interaction. Green color, hydrophobic interaction; purple color, polar interaction. **c** pKi of Clobenpropit, VUF6884 and Clozapine binding of H₄R mutants. **d** pEC₅₀ of Gᵢ activation of H₄R mutants by Clobenpropit, VUF6884 and Clozapine. The green bars indicate agonist and the red bars indicate inverse agonist. From **c**, **d**, data are presented as mean values ± SEM.; $n = 3–4$ independent experiments for **c** and $n = 3–7$ independent experiments for **d**. Each point in the figure represents an independent experiment. Source data are provided as a Source Data file.

imidazole ring of histamine or clobenpropit. The dibenzodiazepine ring is positioned toward T178⁵·⁴² and E182⁵·⁴⁶. Similar to the mutagenesis data observed with clobenpropit, mutant D94³·³²A/Q cannot interact anymore with the two ligands, while mutants in the π-π network significantly reduce receptor binding of VUF6884 and clozapine. Conversely, other mutations have minimal or negligible effects on receptor binding and activation by clozapine and its analog. (Fig. 3c, d).

Interestingly, Q347A increases clozapine affinity and activity, while the effect is minimal with VUF6884. A closer examination of the binding poses of clozapine and VUF6884 shows that Q347 is closer to the dibenzodiazepine ring of clozapine than that of VUF6884 (Supplementary Fig. 4a). Mutation of Q347 to a small residue A (Q347A) may release the clash and accounts for the increase of clozapine activity.

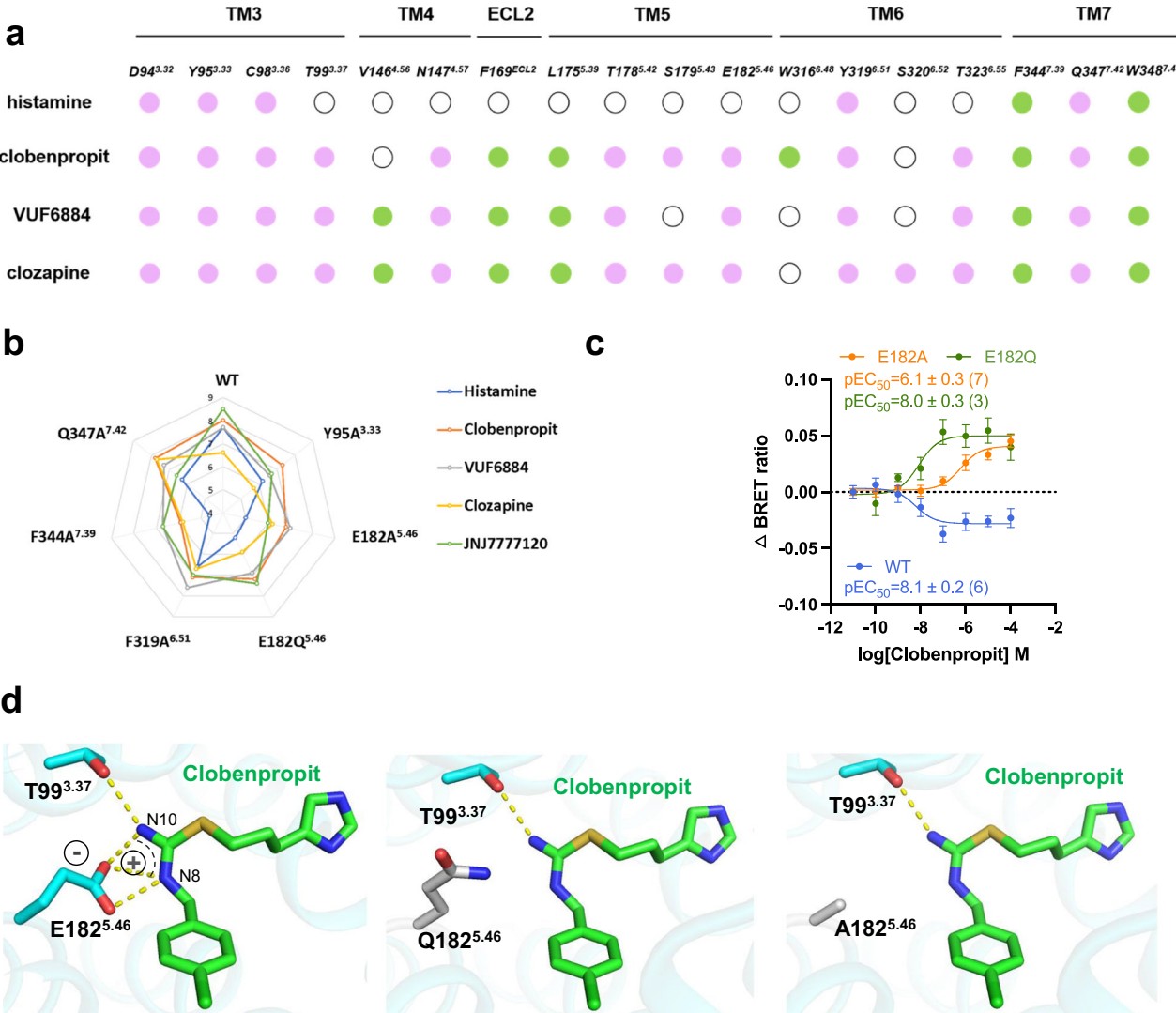

**Fig. 4 | Insights into ligand binding. a** A schematic summary of histamine, Clobenpropit, VUF6884, and Clozapine/receptor interactions. Green solid circle, hydrophobic interaction; purple solid circle, polar interaction; white emptied circle, no interaction. **b** Radar chart for affinities of ligands the four agonists to the wild-type and mutant H4R receptors, as measured by [3H]JNJ7777120 binding.

**c** BRET-based Gi-protein activation assay of the wild-type and E182 H$_4$R mutants. Data are presented as mean values ± SEM.; $n = 6$ independent experiments for WT, $n = 7$ independent experiments for E182A and $n = 3$ for E183Q. Source data are provided as a Source Data file. **d** A structural analysis of the interaction of clobenpropit with E182 and its mutants.

## Insight into H$_4$R ligand recognition and receptor activity

A ligand interaction map of all ligands shows that the binding of histamine mainly involves residues from TM3 and TM7 (only the left half of the pocket), while the binding of the other agonists involves the whole pocket (TM3, TM4, ECL2, TM5, TM6, and TM7) (Fig. 4a). We conducted a comparison between the receptor binding data and the functional assay data regarding receptor activation. The comparison reveals a high degree of consistency between the binding data and the receptor activation data (Supplementary Fig. 10 and Table 2) and only minor differences were noticed. More importantly, in line with the cryo-EM observation, the site-directed mutagenesis studies revealed a distinct pattern for the interaction between various H$_4$R agonists and H$_4$R (Fig. 4b). As can be seen in the spider-web representation, the binding of histamine is most affected by the various mutations, especially the F344$^{6.51}$A and the E182Q$^{5.46}$ and E182A$^{5.46}$ mutations (Fig. 4b). Clozapine seems least affected by the mutations, probably due to its relatively low affinity. The binding of clobenpropit or VUF6684 is also clearly affected by these mutations, but the drop in affinity is not as high as for histamine. Most interestingly, both E182$^{5.46}$ mutants (E182Q$^{5.46}$ and E182A$^{5.46}$) convert the agonist clobenpropit into an

inverse agonist (Fig. 4c and Supplementary Fig. 8b). Upon closer examination of clobenpropit receptor binding, it becomes evident that the negatively charged carboxyl group of E182$^{5.46}$ forms a robust salt-bridge interaction with the positively charged N8 and N10 of the iso-thiourea group (Fig. 4d, left panel), effectively stabilizing the receptor in an active state. Conversely, mutations of the negatively charged E182$^{5.46}$ to a neutral glutamine (Q) or alanine (A) disrupt the salt-bridge interaction, rendering the receptor incapable of maintaining an active state (Fig. 4d, middle and right panel). In fact, the site of E182$^{5.46}$ has also been implicated in playing crucial roles in regulating H$_1$R, H$_2$R, and H$_3$R activities. For instance, N192A$^{5.46}$ totally abolished H$_1$R activity[11], while T190A$^{5.46}$ and E206A$^{5.46}$ have shown the importance for histamine interaction with H$_2$R and H$_3$R, respectively[25,26]. The structural insight into ligand recognition provides a clear explanation for the divergent receptor activities, facilitating the precise design of novel compounds that target H$_4$R.

### Antihistamine design of H$_4$R

The disruption of the salt-bridge interaction between clobenpropit and E182$^{5.46}$, observed in the E182Q$^{5.46}$ mutant, results in the conversion

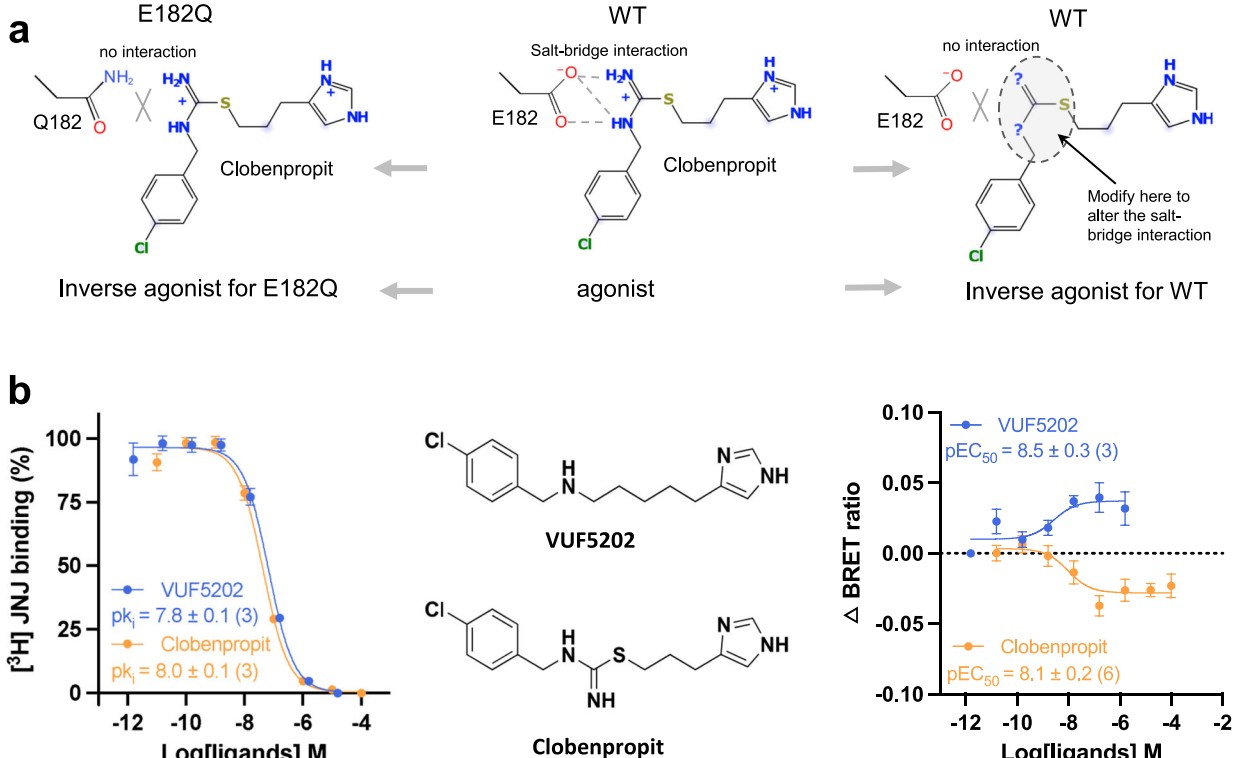

**Fig. 5 | Antihistamine design of H₄R. a** A schematic diagram of the antihistamine design of H₄R, utilizing key information of E182/ligand interaction. **b** VUF5202 exhibits inverse agonist activity. Data are presented as mean values ± SEM.; $n$ = 3 independent experiments for the binding assay (both Clobenpropit and VUF5202), $n$ = 6 for Clobenpropit BRET assay, and $n$ = 3 for VUF5202 BRET assay. Source data are provided as a Source Data file.

of the agonist into an inverse agonist (Fig. 5a, left panel) within the context of the 2 mutant H₄Rs. This intriguing finding raises the question of whether modifying the positively charged N8 and N10 groups of clobenpropit could potentially transform the modified compound into an inverse agonist for the wild-type H₄R (Fig. 5a, right panel), which may have immediate therapeutic potential for associated informatory disease. Guided by this insight, we identified in our library VUF5202, which is differentiated from clobenpropit by a substitution of N10 and S11 with a carbon atom (Fig. 5b, middle panel), as a potential candidate for an inverse agonist for H₄R. Previously, VUF5202 has been shown to act as an antagonist of H₃R[27], but has never been evaluated on H₄R. Remarkably, while VUF5202 exhibits the same pKᵢ as clobenpropit in the binding assay (Fig. 5b, left panel), VUF5202 acts as an inverse agonist for the wild-type (WT) H₄R with a potency in the nanomolar range in the BRET-based G-protein activation assay (Fig. 5b, right panel). This successful identification of a novel inverse agonist for H₄R underscores the precision of our structural analysis and provides a robust foundation for the design of new antihistamines to combat inflammatory diseases.

**Mechanism of H₄R activation**

A comparison of the antagonist PF03654746-bound inactive H₃R[13] with the histamine-bound active H₄R shows that the most notable change is the outward movement of TM6 on the intracellular side in the active H₄R, which allows the αH5 of Gαᵢ to engage the intracellular cavity of the receptor (Fig. 6a), the most common feature of GPCR activation[28]. In addition, we also saw a small outward movement of TM7 on the extracellular side of the receptor. We also compared the histamine-bound active H₄R with the AlphaFold[29] prediction of apo H₄R (inactive). The comparison shows a similar TM6 outward displacement when receptor activation (Supplementary Fig. 4b). In H₁R activation, a "squash to activate and expand to deactivate" was proposed based on the shrinkage of the ligand binding pocket caused by histamine pulling

key residues of TM3, TM5 and TM7 inside[11]. When comparing the extracellular side of histamine-bound H₁R with the active state of H₄R, no evident compression of the ligand binding pocket on H₄R is observed (Supplementary Fig. 3c). Generally, receptor activation of class-A GPCR is mediated by a coordinated movement of conserved motifs such as $C^{6.47}W^{6.48}xP^{6.50}$, $P^{5.50}I^{3.40}F^{6.44}$, $N^{7.49}P^{7.50}xxY^{7.53}$ and $D^{3.49}R^{3.50}Y^{3.51}$. We observe TM7 to spin towards $W316^{6.48}$ of the toggle switch (Fig. 6b) which in turn bends TM6 at the middle to allow the outward movement on the intracellular side. For the $P^{5.50}I^{3.40}F^{6.44}$ motif, a signature movement of $F312^{6.44}$ toward TM5 was observed (Fig. 6c). For the $D^{3.49}R^{3.50}Y^{3.51}$ motif, we observed the movement of $R112^{3.50}$ toward the center of intracellular cavity to allow the tip of the αH5 to engage the receptor (Fig. 6d). For the $N^{7.49}P^{7.50}xxY^{7.53}$ motif, the most dominant movement is the shift of $Y358^{7.53}$ toward TM3, a phenomenon seen in most class A GPCR activations (Fig. 6e).

**G-protein engagement**

H₄R almost exclusively couples to Gᵢ signaling which leads to a decrease of cAMP production and an increase of intracellular $Ca^{2+}$. In the cryo-EM structures, the G-protein engagement is mainly mediated by the insertion of αH5 into the intracellular cavity of H₄R. An analysis of the H₄R-Gαᵢ interaction shows that the hydrophobic interactions between a cluster of hydrophobic residues $L353^{G.H5.25}$, $L348^{G.H5.20}$, $I344^{G.H5.16}$, $I343^{G.H5.15}$ and a patch of the hydrophobic surface formed by $L308^{6.40}$, $L305^{6.37}$, $L301^{6.33}$, $L201^{5.65}$, $I197^{5.61}$ of the H₄R is the main driver (Fig. 7a and Supplementary Fig. 11a). This finding aligns with the analysis of multiple G-protein couplings on ADGRL3[30] and GPR110[31], suggesting that hydrophobic interactions play a crucial role in determining Gᵢ engagement. Interestingly, we also found the ICL2 of H₄R to form extensive polar interaction with Gαᵢ to stabilize the engagement, namely $S115^{3.53}$, $R123^{ICL2}$, and $Q125^{ICL2}/H126^{ICL2}$ of ICL2 form polar interactions with $N347^{G.H5.19}$, $E33^{G.S1.01}$ and $R32^{G.hns1.03}$ of Gαᵢ, respectively (Fig. 7b). When comparing the engagements of H₄R with

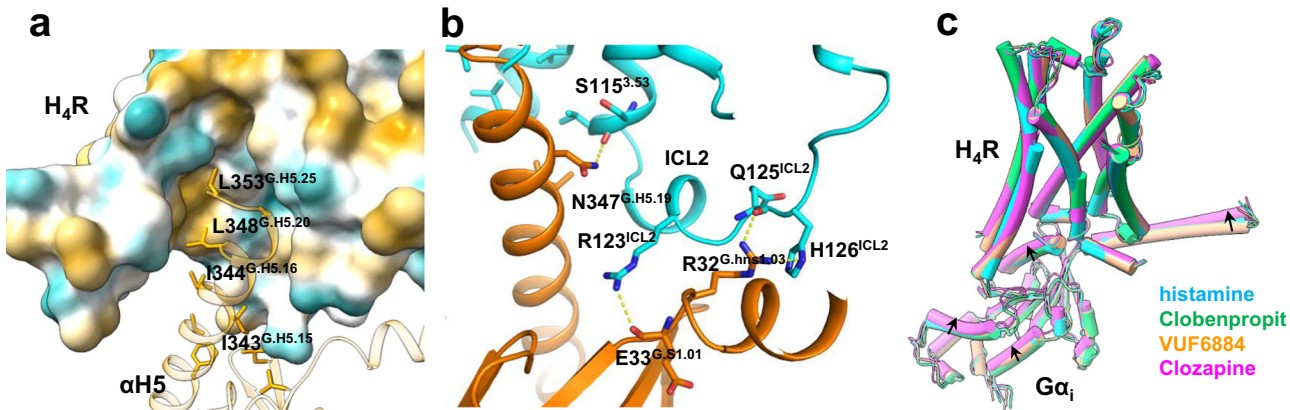

**Fig. 6 | H₄R activation. a** A comparison of the overall structure of the active H₄R (histamine-bound) with the inactive H₃R (PFO3654746-bound, pdb:7f61). **b–e** A comparison of the CWxp, PIF, DRY, and NPxxY motif, respectively, between the active H₄R (histamine-bound) with the inactive H₃R.

**Fig. 7 | The Gᵢ engagement of H₄R. a** The Gᵢ engagement of H₄R is featured by hydrophobic interaction between the αH5 of Gᵢ and the intracellular side of the receptor. **b** The ICL2 of H₄R forms extensive polar interactions with Gαᵢ in the agonist-bound H₄R-Gⁱ complex. **c** A comparison of the overall Gαᵢ conformations of histamine, Clobenpropit, VUF6884, and Clozapine-bound H₄R/Gᵢ complexes.

Gᵢ across different ligands, it is evident that while the overall engagements exhibit a high degree of similarity (r.m.s.d. = 0.603 Å over 1065 pairs of Cα atoms), there are slight differences observed in the clozapine-bound H₄R/Gᵢ complex compared to the histamine-, clobenpropit-, and VUF6884-bound H₄R/Gᵢ complexes (Fig. 7c). The most notable distinction is the 4° outward sway of the α N-terminal helix (αN) and a displacement of αH5 towards the receptor in the Clozapine-bound H₄R/Gᵢ complex (Supplementary Fig. 11b, c).

## Discussion

Histamine receptors have long been recognized as successful targets for treating immune-related disorders and allergies, with anti-histamines against H₁R being widely prescribed for allergy relief[1]. Clinical successes have been also achieved with H₂R and H₃R for respectively the treatment of gastric ulcers and narcolepsy[1]. Despite the lack of clinical success thus far, H₄R holds promise for developing new therapeutic solutions for allergic and inflammatory diseases[1,5]. The

medicinal chemistry field has benefited from the first structure of antagonist-bound $H_1R$ many years ago[10]. Subsequently, the active structure of $H_1R$ provides further insight into the distinct mechanisms by which agonists and antagonists modulate receptor activity[11], thereby bolstering the design and advancement of novel antihistamines targeting $H_1R$.

However, until now, the structural information of $H_4R$ has been absent, and most $H_4R$ antihistamines have been discovered following high throughput screening, fragment-based screening, or successful, classical scaffold hopping strategies[1,5]. Our structural study of $H_4R$ offers new important information for new structure-based approaches. Our data on $H_4R$-$G\alpha_i$ complexes, first of all, reveal a completely different ligand binding mode for histamine compared to its interaction with $H_1R$[11] (Fig. 2d). In $H_1R$, the imidazole of histamine is pointing towards TM3/TM5 side and forms crucial ionic interaction with $N198^{5.46}$, while in $H_4R$, histamine only occupies the pocket on the TM7 side where a π-π network formed by $F344^{7.39}$, $W348^{7.43}$ and $Y319^{6.51}$, leaving the pocket on the TM3/TM5 side empty. Other, larger $H_4R$ agonists, like clobenpropit and clozapine, occupy also the TM3/TM5 side of the binding pocket. We assigned a phosphate molecule to an unidentified density within the histamine binding pocket. MD simulations revealed that the phosphate molecule played a crucial role in stabilizing the binding of histamine to the receptor Concurrently, our investigation does not rule out the possibility that other anions may also play a role in stabilizing histamine binding.

One of the most interesting discoveries of our study is the observation that mutation on the single residue $E182^{5.46}$ converts clobenpropit from a partial agonist into an inverse agonist of $H_4R$. $E182^{5.46}$ is not directly involved in histamine binding, but plays a crucial role in the binding of the other 3 tested agonists (Figs. 2a and 3a, b). Particularly, it forms a salt-bridge interaction with the positively charged N8/N10 of clobenpropit, which positions the ligand to activate the receptor. A slight change of the net charge at this position ($E182^{5.46}$ to $Q182^{5.46}$) without altering the overall conformation causes the loss of the key salt-bridge interaction, resulting in a complete inactivation of the receptor while retaining the affinity of the clobenpropit (Fig. 4b–d). This demonstrates the tight connection between receptor activation of $H_4R$ and subtle changes in ligand binding. The identification of $E182^{5.46}$ as a regulator of receptor activation carries substantial implications for the design of novel antihistamines targeting $H_4R$. The development of new selective $H_4R$ inverse agonists is generally considered of interest for inflammatory and allergic conditions[1]. The discovery of VUF5202 as a novel inverse agonist for $H_4R$ is a proof of concept for this principle for novel antihistamine design for $H_4R$. Moreover, VUF5202 has a completely different chemical scaffold compared to the known $H_4R$ antihistamines JNJ7777120 and Toreforant[32], both of which didn't succeed in clinical trials[9], highlighting the potential of developing innovative antihistamines for $H_4R$ via this new structural insight.

While our manuscript was under review, a similar study reported the histamine- and imetit-bound $H_4R$ in complex with $G_q$[33], an unusual coupling partner of $H_4R$. Compared to this study, we reveal the coupling information with the $G_i$ protein, which is considered the primary transducer of $H_4R$. Moreover, clobenpropit, VUF6884, and Clozapine are large $H_4R$ ligands, having different backbones compared to the small agonist's histamine and imetit. Therefore, our structures could provide additional information for designing novel ligands/modulators targeting $H_4R$.

Collectively, we successfully determined the cryo-electron microscopy (cryo-EM) structures of the $H_4R$/$G_i$ complex in association with 4 different (partial) agonists, unveiling a distinctive mode of histamine binding specific to $H_4R$. Furthermore, we have discovered that $E182^{5.46}$ plays a crucial role in determining ligand efficacy and $H_4R$ activation and discovered VUF5202 as a novel inverse agonist for $H_4R$. Together with the mechanistic insights into GPCR activation and $G_i$

engagement, our study provides a structural basis for the understanding of $H_4R$ signaling and offers a logical foundation for the development of novel antihistamines targeting $H_4R$.

## Methods

### Constructs

The human $H_4R$ gene, optimized for codon usage, was incorporated into the pFastBac1 baculovirus expression vector. The gene sequence included an HA-signal peptide sequence at the N-terminus and a LgBiT fusion at the C-terminus, followed by a Tobacco etch virus (TEV) cleavage site and two maltose-binding protein (MBP) domains. To enhance protein expression and folding, the ICL3 loop of $H_4R$ (residues 215-286) was removed. Additionally, the C-terminal fusion of human $G\beta_1$ with HiBiT[34] was cloned into a separate pFastBac plasmid, as described in the VIP1R paper. The pFastBac plasmid also contained clones of the human dominant-negative $G\alpha_{i1}$ (bearing the G203A/A326S mutant for the histamine- and Clozapine-bound $H_4R$, and the S47N/G203A/A326S/E245A mutant for the Clobenpropit- and VUF6884-bound $H_4R$), wild-type human $G\beta_1$, wild-type human $G\gamma_2$, and the scFv16 encoding the single-chain variable fragment of mAb16, as previously described. Site-directed mutagenesis was performed by polymerase chain reaction (PCR) using the N-terminal HA-epitope tagged human $H_4R$ (GenBank: NM_021624) as a template. PCR products were subcloned into the mammalian expression plasmid pcDEF3 using flanking BamHI and XbaI restriction sites and verified by DNA sequencing.

### Protein expression and purification

To express the proteins, *Spodoptera frugiperda (Sf9)* cells were co-infected with baculoviruses carrying $H_4R$, $G\alpha_{i1}$, $G\beta_1$, $G\gamma_2$, and scFv16 at a ratio of 1:100 (virus volume to cell volume). The cells were harvested 48 h post-infection. The cell pellets were resuspended in a buffer containing 20 mM Hepes, 150 mM NaCl, 10 mM $MgCl_2$, 20 mM KCl, 5 mM $CaCl_2$ at pH 7.5, supplemented with 0.5 mU/mL apyrase, and homogenized by douncing approximately 30 times. Throughout the purification process, ligands including HSM, Clobenpropit, VUF6884, and Clozapine were added at concentrations of 100 μM, 10 μM, 5 μM, and 5 μM, respectively. After incubating the lysate at room temperature for 1 h, 0.5% (w/v) lauryl maltose neopentylglycol (LMNG) and 0.1% (w/v) cholesteryl hemisuccinate TRIS salt (CHS) were added to solubilize the membranes, followed by incubation at 4 °C for 2 h. The lysate was then subjected to ultracentrifugation at $65,000 \times g$ and 4 °C for 40 min. The supernatant was incubated with an amylose column for 2 h, washed with a buffer containing 25 mM Hepes, pH 7.5, 150 mM NaCl, 0.01% LMNG, and 0.002% CHS, and eluted with the same buffer supplemented with 10 mM maltose. The eluate was concentrated and treated with homemade TEV protease overnight at 4 °C. Subsequently, the sample was separated on a Superdex 200 Increase 10/300 GL gel filtration column using a buffer composed of 25 mM Hepes, pH 7.5, 150 mM NaCl, 0.00075% (w/v) LMNG, 0.00025% glyco-diosgenin (GDN), and 0.0002% (w/v) CHS. The peak corresponding to the $H_4R$/ Gi complex was concentrated to approximately 10 mg/mL and snap-frozen for subsequent cryo-EM grid preparation.

### Grid preparation and cryo-EM data collection

A protein complex sample (~10 mg/mL) of approximately 3–5 μL was loaded onto Cu holey carbon grids (Quantifoil R1.2/1.3) that were pre-treated with glow charging (Quantifoil GmbH). The loaded grids were then vitrified by rapidly plunging them into liquid ethane using a Vitrobot Mark IV (Thermo Fisher Scientific). The Vitrobot settings used were as follows: blot force 10, blot time 5 s, humidity 100%, and temperature 4 °C. The prepared grids, containing evenly distributed particles in thin ice, were placed into a FEI 300 kV Titan Krios transmission electron microscope (TEM) equipped with a Gatan Quantum energy filter. Imaging was performed using a Gatan K2 Summit direct electron detector employing a super-resolution counting model, with a pixel size

of 0.55 Å at a magnification of 64,000×. The energy filter slit was adjusted to 20 eV. Each image consisted of 40 frames, with a total exposure time of 7.3 s and a dose rate of 1.5 e/Å$^2$/s (resulting in a total dose of 60 e/Å$^2$). The nominal defocus value ranged from −1.2 to −2.2 μm.

## Data processing
The cryo-electron microscopy (cryo-EM) data were processed using a standard pipeline established in our laboratory[35]. Initially, the raw movies were binned once (1.1 Å) and corrected for motion using MotionCor2[36]. Subsequently, the contrast transfer function (CTF) parameters were estimated using CTFFIND 4.1[37]. Particle picking was performed using crYOLO[38], followed by reference-free 2D classification in RELION[39]. The well-defined 2D features obtained from this classification were used to generate an initial model using cryoSPARC's ab initio method[40]. The generated initial model served as a reference for further refinement steps in RELION. A 3D classification was conducted, resulting in 3–4 classes. The best class, displaying clear secondary structure features, was selected for Non-uniform Refinement in cryoSPARC.Subsequently, a no-alignment 3D classification was performed in RELION, employing 6–10 classes and applying a mask on the complex. Bayesian polishing[41] and additional rounds of Non-uniform Refinement were carried out to enhance the map quality. The resolution of the final map was estimated using the gold standard Fourier Shell Correlation (FSC) criterion at FSC = 0.143. Local resolution estimations were performed using an implemented program in cryoSPARC.

## Model building
We employed AlphaFold prediction[29] of human H$_4$R (AF-Q9H3N8-v1) as initial models to guide the process of model rebuilding against the electron microscopy map. The docking of these models into the density map was performed using UCSF Chimera[42]. Iterative manual adjustments were carried out in Coot to refine the models, followed by Rosetta cryo-EM refinement[43] and Phenix real space refinement[44] to further improve the structural accuracy. For the visualization and preparation of structural figures, UCSF ChimeraX[45] and PyMOL (https://pymol.org/2/) were utilized.

## Molecular docking
The docking methodology employed in this study follows a similar approach to previous research[46]. Initially, the histamine-bound H$_4$R structure was utilized as the starting model and prepared/minimized using established protocols. 3D model files (in SDF format) for the candidate ligands were obtained from PubChem. The candidate ligands were then positioned within the ligand binding pocket using the triangle matcher, with the London docking score used for assessment. Refinement steps were performed utilizing a rigid receptor and GBVI/WSA docking scoring.

## Synthesis of [$^3$H]JNJ7777120
As described early[47], a precursor for radiolabeling, (5-Chloro-1H-indole-2-yl)-(piperazine-1-yl)-methanone$^x$ (0.15 mg, 39.6 μmol) was dissolved in 117 μL of [$^3$H]methyl nosylate (Perkin Elmer, 854 MBq/mL in acetonitrile). The reaction mixture was heated for 15 min at 70 °C. Next, the reaction was allowed to cool to ambient temperature, diluted with 2 mL of HPLC eluent and injected onto preparative HPLC (Jasco PU-2080 Pump, Jasco UV-2075 UV detector (Jasco, Utrecht, The Netherlands) mounted with a Luna C18 10*250 mm, 100 Å, 10 μm column en eluted with 25/75 acetonitrile/water, 0.2% DIPEA at 5 mL/min, UV was measured at 225 nm. The product eluted at 52 to 54 min and was collected in a solution of 60 mL of water. The total mixture was purged over a Sep-Pak tC18 (Waters, Milford, USA) which was pre-washed with 10 mL of ethanol and 20 mL of water, successively. After trapping of [$^3$H]JNJ7777120, the Sep-Pak was washed with 20 mL of water and [$^3$H]JNJ7777120 was obtained with elution of the Sep-Pak with 2 mL of ethanol.

The concentration of [$^3$H]JNJ7777120 in ethanol was determined using beta counting (Hidex 300 SL, Turku, Finland) and found to be 19.7 MBq/mL. The product was analyzed with HPLC (Jasco PU-2080 Pump, Jasco UV-2075 UV detector mounted with a Lablogic (Sheffield, UK) β-RAM Scintilation detector) using a Luna C18, 4.6*250 mm, 100 Å, 5 μm column which was eluted with 35/65 acetonitrile/water, 0.1% DIPEA at 1 mL/min. UV was measured at 225 nm. The radiochemical purity was 97.9% and no chemical impurities were observed. The molar activity was 2.57 MBq/nmol, based on the used [$^3$H]methyl nosylate.

## Radioligand binding experiments
Two million HEK293T cells were seeded in 100 mm tissue-culture dishes in Dulbecco's modified eagle medium (DMEM) supplemented with 10% FBS, penicillin (100 IU/mL), and streptomycin (100 μg/mL) at 37 °C with 5% CO$_2$. The next day, cells were transiently transfected with 2.5 μg DNA encoding for human wild-type HA-H$_4$R or mutant HA-H$_4$R and 2.5 μg empty pcDEF3 using 30 μg 25 kDa linear polyethylenimine. After 48 h, cells were washed and collected with ice-cold phosphate-buffered saline (PBS) and centrifuged at 1900 × $g$ for 10 min at 4 °C. Cell pellets were stored at −20 °C. Next, cell pellets were resuspended in binding assay buffer (50 mM Tris-HCl, pH 7.4) and sonified for 15 s before each experiment. Radioligand competition binding was measured on 50 μL cell homogenates expressing wild-type or mutant H$_4$R using 25 μL [$^3$H] histamine or [$^3$H] JNJ7777120, and 25 μL buffer or unlabeled ligands. Nonspecific radioligand binding was determined in the presence of 10 μM JNJ7777120. After 2 h at 25 °C, the incubations were terminated by rapid filtration over a 0.5% PEI-coated 96-well GF/C filter plate through three rapid wash steps with ice-cold wash buffer (50 mM Tris-HCl, pH 7.4) using a Perkin Elmer 96-well Filtermate-harvester (Perkin Elmer, Groningen, the Netherlands). The GF/C filter plates were dried at 52 °C for 1 h and 25 μL Microscint-O scintillation liquid was added per well. Filter-bound radioactivity was measured using a Microbeta2 plate counter (Perkin Elmer) after a 120 min delay.

Data for competition binding were analyzed by nonlinear regression analysis using GraphPad Prism 9.5.1. IC$_{50}$ values were obtained by fitting the data from the competition studies to a one-site competition model. The Ki of unlabeled ligands was calculated using the Cheng-Prusoff equation with radioligand binding affinity values determined by the homologous displacement equation. Competition binding graphs represented the pooled data from at least three independent experiments performed in duplicate.

## BRET-based Gαi activation assay and anti-HA ELISA
For the BRET-based Gαi activation assay, two million HEK293T cells were seeded in 100 mm tissue-culture dishes. The next day, 1 μg plasmid encoding for wild-type or mutant HA-H$_4$R was transiently cotransfected with 1.5 μg bicistronic plasmid encoding for a BRET-based Gαi sensor[18] using 20 μg 25 kDa linear polyethylenimine. An empty pcDEF3 vector was added to normalize the total amount of DNA to 5 μg per 100 mm dish. At 24 h after transfection, 50,000 cells per well were transferred into 0.01% Poly-L-Lysine (PLL) precoated white and transparent 96-well plates (Greiner, #655083) and further maintained for 24 h at 37 °C with 5% CO$_2$. Cells in the white plates were washed with HBSS and incubated with agonists for H$_4$R and furimazine (Nano-Glo®, Promega). After 40 min at room temperature (RT), luminescence was measured using the CLARIOstar Plus Microplate reader at 535-20 and 470-80 nm. The BRET ratio was determined as the acceptor emission divided by the donor emission. At least three independent experiments were performed in duplicate, and data were normalized to the vehicle using GraphPad Prism 9.5.1. Significant analysis was performed using a one-way AVONA test under the multiple comparisons of Dunnett ($****p < 0.0001, ***p = 0.0002, **p = 0.02, *p = 0.03$).

To measure (mutant) H$_4$R protein expression transfected cells in the transparent plates were washed with TBS buffer (50 mM Tris and 150 mM NaCI) and fixed with 4% PFA for 30 min at RT. Cells were

incubated overnight at 4 °C with anti-HA (Sigma, Cat# 11867423001, diluted 1000-fold from stock), followed by incubation with anti-rat-HRP (diluted 1000-fold from stock) for 2 h at RT. Cells were washed twice between all antibody incubations finally and the absorption at 450 nm was measured using the CLARIOstar Plus Microplate reader after the addition of substrate solution (Mix TMB and $H_2O_2$).

## Molecular dynamics simulation

The cryo-EM structure of histamine-bound $H_4R$ (receptor only) was used as the initial model in the MD simulation. The ICL3 break (204-292) was filled with residues AAGAAA. The model was prepared and parameterized in CHARMM-GUI[48,49]. Protonation states of all titratable residues were assigned at pH 7.0. Histamine was bi-protonated according to a previous report[15]. PO4 was protonated as $HPO4^{-2}$ according to Protonate3D analysis[50]. The $H_4R$ model was inserted into a lipid bilayer containing POPC (palmitoyl-2-oleoyl-sn-glycero-3-phosphocholine) and cholesterol at a 4:1 ratio. The membrane had dimensions of $65 \times 65$ Å, with 22.5 Å of water on the top and bottom (resulting in final system dimensions of approximately 65 x 65 x 120 Å). The ion concentration was set to 0.15 M KCl (see Supplementary Table 1 for the details of the system setting). The Amber force fields were configured as follows: protein FF19SB, lipid LIPID17, water TIP3P, and ligand GAFF2. Simulations were conducted using the Amber20 package[51]. The system underwent initial energy minimization for solvent and all atoms, followed by heating to 300 K over 300 ps and equilibration for 700 ps. Subsequently, three independent production runs of 200 ns each were performed with a time step of 2 fs. During simulations, the Particle Mesh Ewald algorithm calculated long-range electrostatic interactions, while a cutoff of 10 Å was applied for short-range electrostatic and van der Waals interactions. SHAKE algorithm constraints were applied to all bonds involving hydrogens. Temperature (300 K) and pressure (1 atm) were controlled by the Langevin thermostat and Berendsen barostat, respectively. Trajectory analysis and visualization were carried out using VMD[52], and video recording was facilitated by VMD.

## Reporting summary

Further information on research design is available in the Nature Portfolio Reporting Summary linked to this article.

## Data availability

All data produced or analyzed in this study are included in the main text or the Supplementary Figs./tables. Source data are provided in this paper. The cryo-EM density maps and atomic coordinates have been deposited in the Electron Microscopy Data Bank (EMDB) and Protein Data Bank (PDB) under accession numbers EMD-36712 and 8JXT for $H_4R$/Histamine/$G_i$ complex; EMD-36716 and 8JXX for $H_4R$/Clobenpropit/$G_i$ complex; EMD-36715 and 8JXW for $H_4R$/VUF6884/$G_i$ complex and EMD-36714 and 8JXV for $H_4R$/Clozapine/$G_i$ complex. The MD simulation data were deposited to Zenodo (ID: 10802634) Source data are provided in this paper.

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

## Acknowledgements

This work was supported by the Startup Funds of HIT Center for Life Sciences; and the National Natural Science Foundation of China (32070048 to Y.H.). Shuang Shi was supported by a grant from China (CSC grant number 202006310016). R.X. was supported by "the Fundamental Research Funds for the Central Universities".

## Author contributions

R.X. made the constructs, expressed and purified the proteins and assembled the H$_4$R/G-protein complex, prepared and screened the grids, analyzed the data, and prepared the figures. S.S. conducted ligand binding, BRET-based Gi activation, and surface expression experiments, analyzed the data, and prepared the figures. H.F.V. supervised ligand binding experiment, BRET assay, and analyzed data. Z.X., Y.Q., Y.D., J.L. and K.C. cultured the cells and prepared the plasmids. A.W. synthesized and provided [$^3$H]JNJ7777120. A.Z. and C.G. collected cryo-EM data. R.L. designed the experiments, supervised the project, analyzed the data, and wrote the manuscript with Y.H. Y.H. designed experiments, solved the structures, analyzed data, supervised the project, and wrote the manuscript with R.L. All authors contributed to the data interpretation and preparation of the manuscript.

## Competing interests

The authors declare no competing interests.
