## [Peer Review File · Nature Communications]

Structural basis of ligand recognition and design of antihistamines targeting histamine H4 receptorREVIEWER COMMENTS

Reviewer #1 (Remarks to the Author):

This manuscript provides new insight into ligand binding and activation of the histamine H4 receptor using multiple cryo-EM structures. This work adds significantly to the understand of the H4 receptor and to the design of novel ligands. Overall the data appears to be solid and well presented.

A few comments for the authors to consider:

1. One of the most interesting findings in the paper is around role of E182 in the functional activity of clobenpropit. VUF6884 and clozapine also interact with this residue. Is it also important for their functional activity?

2. With respect to the E182 data, it is clear in Figure 4c that the mutations E182A and E182Q shift the function from agonist to inverse agonists with pEC50s of 6.1 and 7.8, respectively. However, this makes the presentation of the data in Figure 3D confusing. The correct EC50s are reported for these mutations, but there is no mention that this is now for inverse agonism. What about for all of the other mutations and other ligands? Do the EC50s represent agonism or inverse agonism? The authors should find a way to better clarify this in the figure since it is current confusing.

3. One other curious finding is the effect of the mutation Q347A on the clozapine activity. This increases the affinity and activity of clozapine quite dramatically (even though the text says the other mutations have minimal effect). This is not as prominent with VUF6884. Does this tell us anything more about receptor activation?

Reviewer #2 (Remarks to the Author):

This manuscript by Xia et al., presents cryo-EM structures of the histamine H4 receptor in complex with four different agonists, and draw important insights into ligand recognition, receptor activation and transducer-coupling. Taking lead from structural insights, the authors have also engineered receptor mutants that exhibits a reversal of ligand pharmacology, and provide a framework for structure-guided ligand discovery. Overall, the experiments are well designed and executed, and the insights provide important advance in the field of histamine receptor biology. Although, a recent study has also reported agonist-H4R-G-protein structure (PMID: 37863901), the current manuscript should be seen as

contemporary study, and the recent publication should not compromise the novelty, interest, and the impact of the current manuscript. I would strongly recommend the publication of this manuscript, and I have only a couple of minor suggestions that the authors should consider during revision.

1. The discussion on JNJ compound is somewhat confusing as it is mentioned first as an antagonist and then as an arrestin-biased agonist. The authors should streamline this discussion better for the readers.
2. The color scheme of bar graphs in Extended Data Fig. 4f should be revised as the individual data points are not visible.
3. Some of the supplemental figure panels are too small to be legible, and they can be improved, for example, by separating them into additional supplemental figures.
4. Authors should consider citing and briefly discussing the other study by Im et al., that was published while this manuscript was under peer review.

Arun K. Shukla, PhD

Reviewer #3 (Remarks to the Author):

This manuscript by He, Leurs, and coworkers reports a structural biology study of a medically important G protein-coupled receptor (GPCR), the histamine H4 receptor (H4R). The subtype of histamine receptors contains four members H1R to H4R, which have been considered therapeutic targets for a range of allergic disorders. While the structures of H1R, H2R, and H3R have been determined, this work fills an important gap by adding four cryo-EM structures of H4R/Gi complexes that are bound with histamine, and synthetic agonists clobenpropit, VUF6884 and clozapine. These findings provide structural information and binding mode, enabling further structure-based drug design and development targeting H4R.

This research likely advances our knowledge of agonist and inverse agonist binding poses and mechanisms. Most importantly, it uncovers the essential role of single residue E1825.46 in H4R activity. E1825.46 mutants convert clobenpropit from an agonist of H4R to an inverse agonist. To explain this result, the authors discover that the negatively charged carboxyl group of E1825.46 and the positively charged N8 and N10 of the isothiourea group can form a robust salt bridge to stabilize H4R in an active state. This finding enriches the mechanism to design an antagonist of H4R. Besides these exploring, the authors also clarify how hydrophobic interactions are involved in G-protein engagement in H4R, based on a structure and interaction perspective.

Overall, the manuscript is well prepared, and the research can be novel and significant. The results reveal valuable structural information for understanding H4R biological activities and provide a

mechanism for designing novel drugs targeting H4R. This is likely to generate impacts in the field of GPCR study and GPCR drug discovery. I recommend publication after minor revision.

Some questions from the reviewer (which may improve clarity of the manuscript):

1. Single residue E1825.46 plays an essential role in forming a salt bridge. What is the role of E1825.46 in other histamine receptors? Is this mechanism conserved in the histamine receptor family?

2. In the part of "Mechanism of H4R activation", is it possible to use an inactive H4R to compare with the active H4R, instead of using inactive H3R? It should be more convincing to do so.

Reviewer #4 (Remarks to the Author):

Xia and coauthors reported cryo-EM structures of H4R/Gi complexes bound with histamine and synthetic agonists clobenpropit, VUF6884 and clozapine. They compared the different binding pose of ligands in H4R and found some interest insights for future drug discovery. However I have some concerns for this current version.

1. The binding pose of histamine in H4R evaluation. I would like suggest the author should perform molecular dynamic simulation to evaluate this distinct binding pose in H4R, please run MD in H3R. In addition, they found a phosphate ion nearby the histamine, did the author test the functional role of a phosphate ion for receptor activity?

2. The novelty of inverse agonist VUF5202. The ligand VUF5202 has been reported as kind of antagonist of H3R. The author has mentioned that H3 and H4R have high similarity.

3. Line 189-190, please add a reference for this statement "clozapine has been found to activate H4R, which might be related to the known side effect of agranulocytosis by clozapine." I would like suggest the author discuss the different binding pose of clozapine in histamine receptors and dopamine receptors.

4. Figure 2a-2b, please show the distance and detailed interaction between the amine group of histamine and surrounding residues. Could the author have predict the stability of histamine in H4R?

REVIEWER COMMENTS

Reviewer #1 (Remarks to the Author):

This manuscript provides new insight into ligand binding and activation of the histamine H4 receptor using multiple cryo-EM structures. This work adds significantly to the understand of the H4 receptor and to the design of novel ligands. Overall the data appears to be solid and well presented.

Thank you very much for your positive comments on our study.

A few comments for the authors to consider:

1. One of the most interesting findings in the paper is around role of E182 in the functional activity of clobenpropit. VUF6884 and clozapine also interact with this residue. Is it also important for their functional activity?

Yes, E182 interacts with clobenpropit, VUF6884 and clozapine. Mutants of E182A and E182Q have minor effect on the binding of these three ligands (Fig. 3c and new Supplementary Fig. S6,S7a. However, in the functional assay, mutants of E182A and E182Q have diverse effects on clobenpropit, VUF6884 and clozapine; where they convert agonist clobenpropit into inverse agonist, while have almost no effects on VUF6884 and clozapine. This may be due to the fact that E182 makes a direct salt-bridge interaction (strong interaction) with clobenpropit, whereas it only loosely associated with the other two ligands (weak interaction).

2. With respect to the E182 data, it is clear in Figure 4c that the mutations E182A and E182Q shift the function from agonist to inverse agonists with pEC50s of 6.1 and 7.8, respectively. However, this makes the presentation of the data in Figure 3D confusing. The correct EC50s are reported for these mutations, but there is no mention that this is now for inverse agonism. What about for all of the other mutations and other ligands? Do the EC50s represent agonism or inverse agonism? The authors should find a way to better clarify this in the figure since it is current confusing.

Yes, it is hard to judge from Fig. 3c whether it is an inverse agonist or not. We are sorry about this. One can judge whether it is agonist or inverse agonist from a full dose-response curves (Fig. 4c and Supplementary Fig. 5), but we agree that in its present form this is a bit confusing. We have plotted agonist curves as green bars and the inverse agonist bars as red bars new Fig. 3.

3. One other curious finding is the effect of the mutation Q347A on the clozapine activity. This increases the affinity and activity of clozapine quite dramatically (even though the text says the other mutations have minimal effect). This is not as prominent with VUF6884. Does this tell us anything more about receptor activation?

We believe that the observed effect for clozapine is affinity-driven. We have now compared

directly the binding poses of clozapine with that of VUF6884. It appears that VUF6884 binds much flatter in the binding pocket, while clozapine bends much more on the dibenzodiazepine ring, and Q347 is much closer to the ring of clozapine than that of VUF6884 and may somehow clash with the ring (Fig. 1 of the rebuttal letter or new Supplementary Fig. S4a). Mutation Q347 to a small residue A (Q347A) may release the clash and accounts for the dramatic increase of clozapine activity, we have discussed this in our main text line 223-227)

Fig. 1 of the rebuttal letter. A comparison of the binding poses of VUF6884 and clozapine in the ligand binding pocket of H4R.

Reviewer #2 (Remarks to the Author):

This manuscript by Xia et al., presents cryo-EM structures of the histamine H4 receptor in complex with four different agonists, and draw important insights into ligand recognition, receptor activation and transducer-coupling. Taking lead from structural insights, the authors have also engineered receptor mutants that exhibits a reversal of ligand pharmacology, and provide a framework for structure-guided ligand discovery. Overall, the experiments are well designed and executed, and the insights provide important advance in the field of histamine receptor biology. Although, a recent study has also reported agonist-H4R-G-protein structure (PMID: 37863901), the current manuscript should be seen as contemporary study, and the recent publication should not compromise the novelty, interest, and the impact of the current manuscript. I would strongly recommend the publication of this manuscript, and I have only a couple of minor suggestions that the authors should consider during revision.

Thank you very much for your positive comments on our study.

1. The discussion on JNJ compound is somewhat confusing as it is mentioned first as an antagonist and then as an arrestin-biased agonist. The authors should streamline this discussion better for the readers.

Sorry for the confusion. JNJ777120 was first generated as a selective antagonist for H4R based on its ability to shut down the Gi activity (Thurmond et al. J Pharmacol Exp Ther. 2004, PMID: 14722321). Later, JNJ777120 was discovered to have the ability to recruit β -arrestin without activating G proteins (Rosethorne et. al. Mol Pharmacol. 2011, PMID: 21134907). In the revised MS (line 69-75), we have rewritten the introduction and hopefully streamlined the discussion more clearly.

2. The color scheme of bar graphs in Extended Data Fig. 4f should be revised as the individual data points are not visible.

Thank you for your suggestion. We have changed this Figure so each individual point can be seen, please see the new supplementary Fig. 7b.

3. Some of the supplemental figure panels are too small to be legible, and they can be improved, for example, by separating them into additional supplemental figures.

We appreciate your suggestion. In response, we have organized Supplementary Fig. 4a-e and 5a-d into new figures, with each figure now containing only two ligand panels. These updated figures, labeled as Supplementary Fig. 5-9, significantly enhance the overall readability.

4. Authors should consider citing and briefly discussing the other study by Im et al., that was published while this manuscript was under peer review.

We fully agree with the reviewer and have added a discussion on this (line 365-371).

Reviewer #3 (Remarks to the Author):

This manuscript by He, Leurs, and coworkers reports a structural biology study of a medically important G protein-coupled receptor (GPCR), the histamine H4 receptor (H4R). The subtype of histamine receptors contains four members H1R to H4R, which have been considered therapeutic targets for a range of allergic disorders. While the structures of H1R, H2R, and H3R have been determined, this work fills an important gap by adding four cryo-EM structures of H4R/Gi complexes that are bound with histamine, and synthetic agonists clobenpropit, VUF6884 and clozapine. These findings provide structural information and binding mode, enabling further structure-based drug design and development targeting H4R.

This research likely advances our knowledge of agonist and inverse agonist binding poses and mechanisms. Most importantly, it uncovers the essential role of single residue E1825.46 in H4R activity. E1825.46 mutants convert clobenpropit from an agonist of H4R to an inverse agonist. To explain this result, the authors discover that the negatively charged carboxyl group of E1825.46 and the positively charged N8 and N10 of the isothioureia group can form a robust salt bridge to stabilize H4R in an active state. This finding enriches the mechanism to design an antagonist of H4R. Besides these exploring, the authors also clarify how hydrophobic interactions are involved in G-protein engagement in H4R, based on a structure and interaction perspective.

Overall, the manuscript is well prepared, and the research can be novel and significant. The results reveal valuable structural information for understanding H4R biological activities and provide a mechanism for designing novel drugs targeting H4R. This is likely to generate impacts in the field of GPCR study and GPCR drug discovery. I recommend publication after minor revision.

Thank you very much for your positive comments on our study.

Some questions from the reviewer (which may improve clarity of the manuscript):

1. Single residue E1825.46 plays an essential role in forming a salt bridge. What is the role of E1825.46 in other histamine receptors? Is this mechanism conserved in the histamine receptor family?

Indeed the position 5.46 in TM5 is important in the histamine receptor family. Our previous data on H1R shows that N198^{5.46} directly form hydrogen-bond with the N τ of histamine and the N198A mutant completely abolished receptor activity (Xia et. al. NC 2021, PMID: 33828102). Also in H2R and H3R, mutagenesis studies of respectively T190A^{5.46} and E206A^{5.46} have shown the importance of these residues for the interaction with histamine. In the revised MS we refer to those studies and we have added the appropriate references (line 251-254).

H3R: TM5 mutation E206A (5.46)

Uveges, A. J., Kowal, D., Zhang, Y., Spangler, T. B., Dunlop, J., Semus, S., & Jones, P. G. (2002). The role of transmembrane helix 5 in agonist binding to the human H3 receptor. *The Journal of pharmacology and experimental therapeutics*, 301(2), 451–458.

H2R: TM5 mutation T190A (5.46)

Gantz, I., DelValle, J., Wang, L. D., Tashiro, T., Munzert, G., Guo, Y. J., Konda, Y., & Yamada, T. (1992). Molecular basis for the interaction of histamine with the histamine H2 receptor. *The Journal of biological chemistry*, 267(29), 20840–20843.

2. In the part of “Mechanism of H4R activation”, is it possible to use an inactive H4R to compare with the active H4R, instead of using inactive H3R? It should be more convincing to do so.

We made an effort to get apo H4R structure, however, it is very challenge to obtain the apo GPCR structure in a short of time. The small size of GPCR prevents it to be directly solved by cryo-EM. The fusion strategy takes time and needs to be optimized for every specific receptor, therefore we opted not to obtain an apo-structure of H4R. The Alpha-fold predictions of most GPCR are based on the inactive state of receptor, and the backbone of AlphaFold prediction is very close to the experimental structure. Therefore, we compared the histamine-bound H4R with the ApphaFold prediction of inactive apo H4R. The comparison shows a similar outward movement of TM6 when receptor activation (new Supplementary Fig. 4b).

Reviewer #4 (Remarks to the Author):

Xia and coauthors reported cryo-EM structures of H4R/Gi complexes bound with histamine and synthetic agonists clobenpropit, VUF6884 and clozapine. They compared the different binding pose of ligands in H4R and found some interest insights for future drug discovery. However I have some concerns for this current version.

Thank you for your positive comments.

1. The binding pose of histamine in H4R evaluation. I would like suggest the author should perform molecular dynamic simulation to evaluate this distinct binding pose in H4R, please run MD in H3R. In addition, they found a phosphate ion nearby the histamine, did the author test the functional role of a phosphate ion for receptor activity?

We run MD simulations on H4R. The triplicated 200ns runs show that histamine and the anion phosphate are very stable during the simulations (Fig. 2 of rebuttal letter or Supplementary Fig. 4c, 4d and movies 1). Since there is no active structure of H3R available, we opted not to conduct a MD simulation on the inactive crystal structure of antagonist-bound H3R.

Addressing the functional role of the anion phosphate proves challenging, given the ubiquitous presence of phosphate in cells, making its elimination in living cells extremely difficult, if not impossible. Consequently, we opted for MD simulations to investigate the impact of phosphate on histamine binding.

A close examination of snapshot from the MD simulations reveals that the phosphate molecule sits at the gap between E182 and Y318, acting as a barrier that prevents histamine from escaping the cage formed by Y318, F344, W348, D94, and Y95 (Fig. 2a of rebuttal letter or Supplementary Fig. 4c). This arrangement leads to a highly stable histamine binding pose, as evidenced by minimal changes during the simulation (left upper panel of Fig. 2b of rebuttal letter or Supplementary Fig.4d), further supported by the straight line from the RMSD analysis (left lower panel of Fig. 2b of rebuttal letter or Supplementary Fig. 4d).

In contrast, simulations without phosphate result in histamine flipping around the binding pocket, as depicted in the snapshots of the simulation (Fig. 2b of rebuttal letter or Supplementary Fig. 4d, right upper panel). This dynamic behavior is reflected in the substantial fluctuation of the RMSD curves (the lower right panel of Fig. 2b of rebuttal letter or Supplementary Fig. 4d). Collectively, these findings suggest that the anion group of phosphate plays a crucial role in stabilizing histamine binding. We also added a discussion of the phosphate in the discussion.

Fig. 2 of rebuttal letter. MD simulations of histamine binding of H4R. a, A snapshot of histamine and phosphate group co-binding of H4R. b, The trajectory analysis of MD simulations of histamine binding with or without the anion PO4 for H4R. The upper panels are snapshots of histamine binding with PO4 (left) and without PO4 (right) in sequential order during the 200 ns simulations. The lower panels are RMSD analysis of histamine during the 200 ns simulations.

2. The novelty of inverse agonist VUF5202. The ligand VUF5202 has been reported as kind of antagonist of H3R. The author has mentioned that H3 and H4R have high similarity.

As mentioned, VUF5202 was taken from our historical library of H₃R compounds. So, indeed the compound itself is not new and we have added the citation to the relevant H₃R paper in the revised paper. Yet, so far we had never evaluated the compound for its function at the H₄R and the observed inverse agonism at the WT H₄R nicely corroborates our findings with the efficacy switch of clobenpropit on the WT and mutant receptors.

Govoni, M., Lim, H. D., El-Atmioui, D., Menge, W. M., Timmerman, H., Bakker, R. A., Leurs, R., & De Esch, I. J. (2006). A chemical switch for the modulation of the functional activity of higher homologues of histamine on the human histamine H₃ receptor: effect of various substitutions at the primary amino function. *Journal of medicinal chemistry*, 49(8), 2549–2557.

3. Line 189-190, please add a reference for this statement “clozapine has been found to activate H4R, which might be related to the known side effect of agranulocytosis by clozapine.” I would like suggest the author discuss the different binding pose of clozapine in histamine receptors and dopamine receptors.

We added the reference

Clozapine activates H4R: Liu, C., Ma, X., Jiang, X., Wilson, S. J., Hofstra, C. L., Blevitt, J., Pyati, J., Li, X., Chai, W., Carruthers, N., & Lovenberg, T. W. (2001). Cloning and pharmacological characterization of a fourth histamine receptor (H(4)) expressed in bone marrow. *Molecular pharmacology*, 59(3), 420–426. <https://doi.org/10.1124/mol.59.3.420>

Cloz/agranulocytosis: Oloyede, E., Blackman, G., Whiskey, E., Bachmann, C., Dzahini, O., Shergill, S., Taylor, D., McGuire, P., & MacCabe, J. (2022). Clozapine haematological monitoring for neutropenia: a global perspective. *Epidemiology and psychiatric sciences*, 31, e83.

We searched the PDB but could not find a clozapine-bound dopamine receptor structure; the closest we can find is the DCZ-bound miniGo-coupled hM4Di (PDB: 8e9x). A comparison of these two receptors shows that H₄R binding pose of clozapine is similar to DCZ in the hM4Di protein (Fig. 3 of the rebuttal letter).

Fig.3 of the rebuttal letter. A comparison of the binding poses of clozapine analogue between H4R and hM4Di (PDB: 8e9x).

4. Figure 2a-2b, please show the distance and detailed interaction between the amine group of histamine and surrounding residues. Could the author have predicted the stability of histamine in H4R?

We have added the distance in Fig. 2a-b. The MD simulation shows that histamine is stable in the ligand binding pocket (Fig. 2 of rebuttal letter or Supplementary Fig. 4c, 4d).

REVIEWERS' COMMENTS

Reviewer #1 (Remarks to the Author):

All previous concerns have been addressed with revised manuscript.

Reviewer #2 (Remarks to the Author):

The authors have satisfactorily addressed the comments made on the original manuscript. I recommend publication of the revised manuscript.

Arun K. Shukla, PhD

Reviewer #3 (Remarks to the Author):

I feel the comments have been properly addressed. A newly published work in Nat. Comm. can be relevant (Nature Communications volume 15, Article number: 84 (2024)), and I recommend citing it. Overall, I recommend publication.

Reviewer #4 (Remarks to the Author):

The authors have addressed my concerns.

Reviewer #5 (Remarks to the Author):

As per the Editor's request, this review only concerns the MD part which was added upon request of another reviewer (and I concur with their opinion). I will not comment on the rest of the manuscript, which received reviews in the previous rounds.

The MD setup is described in its own methods paragraph. It has been conducted on the basis of the experimental structure (assuming it is of sufficient resolution) and CHARMM-GUI, using modern

forcefields that should provide sufficiently solid grounds. The equilibration was relatively short (1 ns total) as was the production run (200 ns), but SHOULD be on the timescales for at least a qualitative stability analysis. Of course, more is better, but it's hard to draw a line. Definitely the authors should be commended for doing the simulations in triplicate. Also, protonation was done for all residues without regard to their local (pKa) environment, which may be questionable, but one would need the resolved structure to check in detail.

My main reservation consists in the fact that the simulation files (trajectories and, if possible, PRMTOP topologies) should be made available. Simulations quickly become obsolete due to missing details, but topology files may help future researchers to reproduce the same or similar results. For example, that www.gpcrmd.org provides GPCR-specific facilities for MD trajectories; but any other form of hosting (or SI attachment) would be acceptable.

This said, I don't have further reservations to publishing.

REVIEWERS' COMMENTS

Reviewer #1 (Remarks to the Author):

All previous concerns have been addressed with revised manuscript.

Thank you.

Reviewer #2 (Remarks to the Author):

The authors have satisfactorily addressed the comments made on the original manuscript. I recommend publication of the revised manuscript.

Thank you.

Reviewer #3 (Remarks to the Author):

I feel the comments have been properly addressed. A newly published work in Nat. Comm. can be relevant (Nature Communications volume 15, Article number: 84 (2024)), and I recommend citing it. Overall, I recommend publication.

Thank you. We cited the paper in the introduction part (line 89).

Reviewer #4 (Remarks to the Author):

The authors have addressed my concerns.

Thank you.

Reviewer #5 (Remarks to the Author):

As per the Editor's request, this review only concerns the MD part which was added upon request of another reviewer (and I concur with their opinion). I will not comment on the rest of the manuscript, which received reviews in the previous rounds.

The MD setup is described in its own methods paragraph. It has been conducted on the basis of the experimental structure (assuming it is of sufficient resolution) and CHARMM-GUI, using modern forcefields that should provide sufficiently solid grounds. The equilibration was relatively short (1 ns total) as was the production run (200 ns), but SHOULD be on the timescales for at least a qualitative stability analysis. Of course, more is better, but it's hard to draw a line. Definitely the authors should be commended for doing the simulations in triplicate. Also,

protonation was done for all residues without regard to their local (pKa) environment, which may be questionable, but one would need the resolved structure to check in detail.

Thank you. We prepared the protein in MOE where residue's local (pKa) environment is considered.

My main reservation consists in the fact that the simulation files (trajectories and, if possible, PRMTOP topologies) should be made available. Simulations quickly become obsolete due to missing details, but topology files may help future researchers to reproduce the same or similar results. For example, that www.gpcrmd.org provides GPCR-specific facilities for MD trajectories; but any other form of hosting (or SI attachment) would be acceptable.

We tried to deposit the MD data to www.gpcrmd.org, however, the site now limits or stops accepting new data deposit, so we deposited our simulation data (trajectories and topologies) into Zenodo (ID: 10802634) [<https://doi.org/10.5281/zenodo.10802634>], and have released the data to public.

This said, I don't have further reservations to publishing.

Thank you.